# Archaeosomes for Oral Drug Delivery: From Continuous Microfluidics Production to Powdered Formulations

**DOI:** 10.3390/pharmaceutics16060694

**Published:** 2024-05-23

**Authors:** Ivan Vidakovic, Karin Kornmueller, Daniela Fiedler, Johannes Khinast, Eleonore Fröhlich, Gerd Leitinger, Christina Horn, Julian Quehenberger, Oliver Spadiut, Ruth Prassl

**Affiliations:** 1Division of Medical Physics and Biophysics, Gottfried Schatz Research Center for Cell Signaling, Metabolism and Aging, Medical University of Graz, 8010 Graz, Austria; ivan.vidakovic@medunigraz.at (I.V.); karin.kornmueller@medunigraz.at (K.K.); 2Institute of Process and Particle Engineering, Graz University of Technology, 8010 Graz, Austria; daniela.fiedler@miex.cc; 3Research Center Pharmaceutical Engineering, 8010 Graz, Austria; khinast@tugraz.at; 4Center for Medical Research, Medical University of Graz, 8010 Graz, Austria; eleonore.froehlich@medunigraz.at; 5Division of Cell Biology, Histology and Embryology, Gottfried Schatz Research Center for Cell Signaling, Metabolism and Aging, Medical University of Graz, 8010 Graz, Austria; gerd.leitinger@medunigraz.at; 6NovoArc GmbH, 1120 Vienna, Austria; christina.horn@novoarc.at (C.H.); julian.quehenberger@tuwien.ac.at (J.Q.); 7Institute of Chemical, Environmental and Bioscience Engineering, TU Wien, 1060 Vienna, Austria; oliver.spadiut@tuwien.ac.at

**Keywords:** archaeosomes, archaeal lipids, oral drug delivery, insulin, dry powder formulation, solid dosage form

## Abstract

Archaeosomes were manufactured from natural archaeal lipids by a microfluidics-assisted single-step production method utilizing a mixture of di- and tetraether lipids extracted from *Sulfolobus acidocaldarius.* The primary aim of this study was to investigate the exceptional stability of archaeosomes as potential carriers for oral drug delivery, with a focus on powdered formulations. The archaeosomes were negatively charged with a size of approximately 100 nm and a low polydispersity index. To assess their suitability for oral delivery, the archaeosomes were loaded with two model drugs: calcein, a fluorescent compound, and insulin, a peptide hormone. The archaeosomes demonstrated high stability in simulated intestinal fluids, with only 5% of the encapsulated compounds being released after 24 h, regardless of the presence of degrading enzymes or extremely acidic pH values such as those found in the stomach. In a co-culture cell model system mimicking the intestinal barrier, the archaeosomes showed strong adhesion to the cell membranes, facilitating a slow release of contents. The archaeosomes were loaded with insulin in a single-step procedure achieving an encapsulation efficiency of approximately 35%. These particles have been exposed to extreme manufacturing temperatures during freeze-drying and spray-drying processes, demonstrating remarkable resilience under these harsh conditions. The fabrication of stable dry powder formulations of archaeosomes represents a promising advancement toward the development of solid dosage forms for oral delivery of biological drugs.

## 1. Introduction

Archaea are a group of single-celled prokaryotic microorganisms. In the phylogenetic tree, they represent—besides bacteria and eukaryotes—the third domain of life. Concerning their ancient evolutionary history, archaea can typically withstand extreme conditions such as high temperatures, acidic pH values, elevated pressure, or high salinity [1]. For example, *Sulfolobus acidocaldarius*, one representative of the thermophilic subgroup of archaea commonly occurring near hydrothermal sources, can endure temperatures as high as 80 °C and acidic pH values as low as 2 [2]. Figure 1a,b show representative transmission electron microscopic (TEM) and scanning electron microscopic (SEM) micrographs of *Sulfolobus acidocaldarius* cells, which were used for the isolation of archaeal lipids applied in this study. The exceptional resistance of archaea against high temperatures and acidic conditions stems from their unique plasma membrane composition, characterized by two main components: diether lipids (DELs) and tetraether lipids (TELs) [3,4]. As their names imply, these lipids contain ether bonds connecting prevalent isoprenoid sidechains to glycerol. This is in contrast to ester bonds in phospholipids and amide bonds found in sphingolipids connecting non-branched fatty acid chains to glycerol in bacterial and eukaryotic membranes. The structures of the most prominent ether lipids extracted from cultivated *Sulfolobus acidocaldarius* are shown in Figure 1c,d.

Similar to conventional phospholipid molecules isolated from eukaryotes, ether lipids extracted from archaea form vesicular structures in aqueous solution [5]. Diether-based archaeal lipids are known to build less permeable and more condensed lipid bilayer membranes compared to those found for conventional diester-based eukaryotic lipids. Tetraether lipids are membrane-spanning lipids that form a monolayer structure that is densely packed due to the presence of the isoprenoid methyl groups and rigid saturated hydrocarbon chains [6,7,8]. Owing to their unique chemical structures and physical characteristics, archaeal lipids can self-assemble into highly stable and stress-resistant liposome-like particles, known as archaeosomes. Concerning structural features, the monolayer-forming tetraether lipids are believed to enhance the membrane packing density, making archaeosomes more stable and less permeable for ions and compounds compared to conventional liposomes [9,10,11]. Thus, archaeosomes are extensively investigated as drug delivery systems for proteins, peptides, gene delivery, or delivery of natural antioxidant compounds [12,13]. In this context, archaeosomes are considered particularly promising for oral drug delivery. When ingested, the compounds encounter a similar hostile environment present in the gastric system of the human body as the archaea face in their natural habitat, not in terms of high temperatures but regarding the acidic pH value and degrading enzymes present in the gastrointestinal (GI) tract. Likewise, archaeosomes are also stable in the presence of bile salts and lipases [12]. While some studies report on the use of archaeosomes as oral delivery systems for various drug molecules [12,13,14], limited information exists on how archaeosomes behave under different physiological conditions, interact with cells mimicking the GI tract, and whether archaeosomes, due to their pronounced stability, can be transformed into a solid dosage form. Dry powders can, for example, be produced by lyophilization [15,16] or spray-drying processes as already shown for liposomes or solid lipid nanoparticles [17,18]. However, heat and shear stress during manufacturing may lead to instability, enhanced particle aggregation, and rapid drug release, which limits pharmaceutical production on a large scale.

This study aimed to establish a continuous manufacturing process of archaeosomes using isolated archaeal lipids from *Sulfolobus acidocaldarius*. The developed preparation process involves a microfluidics-controlled system enabling the concurrent production of archaeosomes and drug loading within a single-step procedure. Two model compounds were chosen for drug encapsulation. One was calcein, a small inorganic molecule serving as a fluorescent marker to estimate the release behavior of encapsulated compounds from archaeosomes. The second one was human insulin, a pharmaceutically relevant example of a peptide drug molecule. The drug-loaded archaeosomes underwent thorough characterization in terms of particle size, zeta potential, membrane structure, and morphology. The stability of archaeosomes was evaluated in both simulated gastric and intestinal fluid. Loading efficiency and release behavior were determined. The interaction and adhesion behavior with the intestinal barrier were evaluated in a trans-well system using a Caco-2 and HT29-MTX co-culture system as a model for the intestinal epithelium. Finally, freeze-drying and spray-drying procedures were implemented to create a solid dosage form.

## 2. Materials and Methods

### 2.1. Materials

1,2-Dioleoyl-sn-glycero-3-phosphoethanolamine-*N*-(lissamine rhodamine B sulfonyl) ammonium salt (DOPE-rhodamine), (*p*-*tert*-Octylphenoxy)polyethoxyethanol (Triton X-100), cholesterol, fluorescein-bis-(methyliminodiacetic acid) (calcein), paraformaldehyde, glutaraldehyde, pancreatin, pepsin, and human insulin were obtained from Sigma^®^Aldrich Handels GmbH, Vienna, Austria. Sodium cacodylate was purchased from Science Services GmbH (München, Germany). Caco-2 cells (ACC169, HTB-37 clone from the German Collection of Microorganisms and Cell Cultures) were provided by E. Fröhlich (Medical University of Graz, Austria) and HT29-MTX-E12 cells were kindly provided by T. Lesuffleur (INSERM UMR S 938, Paris, France). All components used for the in vitro experiments were purchased from ThermoFisher Scientific (Waltham, MA, USA): high-glucose Dulbecco’s modified Eagle medium (DMEM), fetal bovine serum (FBS), a mixture of penicillin and streptomycin (Pen-Strep), non-essential amino acids, TrypLe as the synthetic substitution for trypsin, as well as the fluorescent stains Phalloidin Alexa Fluor 488, Phalloidin Rhodamine 565, and Hoechst 33258. Fluorescence mounting medium DAKO was purchased from Agilent (Santa Clara, CA, USA). Phosphate buffered saline (PBS; pH 7.4, ionic strength of 162.7 mM) was purchased from Gibco, Life Technologies Corporation (Painsley, UK). Translucent membranes with pores of 3 µm in 12-well plates were obtained from Greiner Bio-One (Kremsmünster, Austria). The following filters and membranes were used in the production process: 20 nm Whatman Anotop 25 syringe filters and polycarbonate membranes of various pore sizes purchased from Cytiva Europe GmbH (Vienna, Austria) and Membra-cel dialysis membranes (Viskase, Lombard, IL, USA). All other chemicals such as dimethylsulfoxide (DMSO), 2-propanol (iPA), trichloromethane (chloroform), methanol, absolute ethanol, and HEPES buffer were obtained from Carl Roth GmbH (Karlsruhe, Germany).

Archaeosomes were produced using mixtures of diether and tetraether lipids that were directly isolated from *S. acidocaldarius* [19] and kindly provided in the form of a crude lipid extract by NovoArc GmbH (Vienna, Austria).

### 2.2. Methods

#### 2.2.1. Particle Preparation Using Microfluidics with Simultaneous Loading of Calcein or Insulin

The crude extract containing a DEL:TEL mixture isolated from the archaea was dissolved either in DMSO: iPA (2:1 vol/vol) or absolute ethanol at a total lipid concentration of 20–21 mg/mL depending on the individual batches. Before microfluidic mixing, the suspension was filtered through a 200 nm syringe filter to remove any solid residues that could obstruct the channels of the microfluidics cartridge. As aqueous phase, 10 mM HEPES buffer pH 7.4 (filtered through a 20 nm syringe filter) was used. A single step production of archaeosomes was performed with a NanoAssemblr^®^ microfluidics mixing device (Precision Nanosystems, Vancouver, BC, Canada) using a staggered herringbone mixer [20,21,22,23]. Process parameters were optimized and set as follows: flow rate ratio 2:1 (aqueous to organic phase) and a total flow rate of 1.7 mL/min. This setup resulted in a final lipid concentration of 7 mg/mL.

For drug encapsulation, calcein, which is a green fluorescent marker with the ability to self-quench at higher concentrations, was used [24,25]. To achieve the quenching effect in our preparations the concentration of calcein was set to 60 mM, corresponding to 37.35 mg/mL calcein dissolved in 10 mM HEPES buffer (pH 7.4). The calcein solution was used as an aqueous phase in the microfluidics system. The process parameters were kept constant as described before. Removal of non-encapsulated calcein along with the organic solvent was performed by dialysis with a 14 kDa membrane tube in two steps against 10 mM PBS (0.15 M NaCl, pH 7.4) buffer. An exchange with a buffer of higher salt concentration was necessary to counteract sample dilution due to the high hypertonic effect of dissolved calcein during dialysis.

The second model drug was the human peptide insulin. In this case, the aqueous phase contained an insulin solution in 10 mM HEPES adjusted to a pH of 2.0 using 1 M hydrochloric acid [26]. Due to the incompatibility between insulin and DMSO the lipids were dissolved in absolute ethanol. Retaining all the process parameters as described before, the final insulin concentration in the prepared formulations was approximately 3 mg/mL. Non-encapsulated insulin was rinsed twice with prefiltered 10 mM HEPES buffer (pH 7.4) within a Satorius Vivaspin 20 (Satorius Lab Instruments GmbH, Göttingen, Germany) centrifuge tube with a 100 kDa cut-off membrane. The encapsulation efficiency (EE%) of insulin was determined using the bicinchoninic acid (BCA) protein assay (BCA™Protein Assay, Thermo Scientifc, Rockford, IL, USA) according to the manufacturer’s instructions. Briefly, sample dilutions were pipetted in triplicates together with an albumin solution of known concentration as standard in a 96-well plate format. After the addition of the reagent mixture and incubation at 37 °C, absorbance measurement at 562 nm of the created colored product was performed using the CLARIOstar plate reader (BMG LABTECH GmbH, Ortenberg, Germany). To verify insulin loading into archaeosomes a native gel electrophoresis using a native acrylamide gradient gel (4–12%) was performed in a Mini-ProteanTetra vertical electrophoresis cell (Bio-Rad Laboratories, Feldkirchen, Germany). The gel was stained with Coomassie blue staining solution for 40 min and destained overnight. Imaging of the stained gels was performed using the Gel Doc EZ Imager (Bio-Rad Laboratories, Feldkirchen, Germany).

#### 2.2.2. Particle Size Analysis

To determine the particle size, photon correlation spectroscopy (PCS) was performed using a Zetasizer HS3000 (Malvern Panalytical, Malvern, Worcestershire, UK) which operates with a 10 mW helium–neon laser at a wavelength of 633 nm. The scattered light was measured at an angle of 90°. All samples were previously diluted with buffer to a concentration of approximately 0.3 mg/mL lipids and measured in a quartz cuvette (Hellma GmbH, Müllheim, Germany) at room temperature. The distribution width and the homogeneity of the particle preparation are given by a polydispersity index (PDI) ranging from 0 (for totally homogeneous samples) to 1 (for highly heterogeneous mixtures).

#### 2.2.3. Zeta Potential

To estimate the surface charge of the archaeosomes, the zeta potential was measured using a Zetasizer NANO ZS (Malvern Panalytical, Malvern, Worcestershire, UK). The samples were diluted 1:10 with distilled water (previously filtered through a 20 nm syringe filter) and measured at room temperature in a special cuvette with a folded capillary cell fitted with the electrodes.

#### 2.2.4. Small-Angle X-ray Scattering (SAXS)

To obtain information on the lipid layer organization in terms of layer thickness and lamellarity, SAXS experiments were performed with a SAXSpace system (Anton Paar, Graz, Austria), equipped with a 30 W-Genix 3D microfocus X-ray generator (Xenocs, Sassenage, France) with a Cu anode and an Eiger R 1 M detector system (Dectris, Baden-Daettwil, Switzerland). SAXS patterns were recorded in a q-range from 0.2 to 6.0 nm^−1^, where q = (4π sin θ)/λ is the scattering vector, 2θ the scattering angle, and λ = 0.154 nm the wavelength of the X-ray beam. The data were buffer background corrected, normalized, and corrected for slit collimation geometry. The electron density profile was calculated with the global analysis program (GAP) [27,28], in which ρ represents the local electron density at the distance z from the center plane in the middle of the lipid layer. From the position of the Gaussian modeling of the electron dense lipid headgroup regions, a direct measure for the membrane thickness can be derived defining the head-to-headgroup distance. SAXS measurements were performed at 25 °C with an exposure time of 1.800 s at an archaeosome concentration of 30 mg/mL.

#### 2.2.5. Transmission Electron Microscopy (TEM)

For negative staining TEM, either archaea or solutions of archaeosomes at a concentration of 0.07 mg/mL were dropped onto a glow-discharged (PELCO easiGlow, Plano GmbH, Wetzlar, Germany) carbon-coated copper grid (EMResolutions, C400Cu100, mesh 400), blotted, and stained for 1 min with 1% (*w*/*w*) uranyl acetate solution. Imaging was performed using a Fei Tecnai G^2^ 20 transmission electron microscope (Eindhoven, The Netherlands), operating at an acceleration voltage of 120 kV. Digital images were captured with a Gatan US1000 CCD camera at 2 K × 2 K resolution using the Digital Micrograph software (Version 1.93.1362, Gatan Inc., Pleasanton, CA, USA) [29].

#### 2.2.6. Scanning Electron Microscopy (SEM)

Archaeosomes or archaea samples were pipetted onto a poly-L-lysine-coated glass slide. After incubation, the samples were gently rinsed with water and immediately fixed using a mixture of 2% paraformaldehyde, 2.5% glutaraldehyde, and 0.1 M cacodylate buffer (pH 7.4). After an incubation period of 30 min, the fixative solution was exchanged with cacodylate buffer for another 30 min, followed by a dehydration step using a series of ethanol to remove water from the sample. The ethanol concentration was increased stepwise with short incubation periods in between. At the very end, the 100% ethanol solution was exchanged with pure acetone and the samples underwent a critical point drying step. In another set of samples, the spray-dried and lyophilized powders containing archaeosomes were deposited onto an SEM holder (Plano G301) covered with adhesive carbon tape (Plano G3347). Liquid conductive silver was applied on the rim of each sample holder and dried for several minutes. Both sets of samples were then sputter coated using a BalTec SCD 500 sputter coater (Balzers, Liechtenstein). Imaging was performed using a Zeiss Sigma 500 VP scanning electron microscope (Zeiss, Oberkochen, Germany) operated at 2 kV, using an Everhart–Thornley secondary and backscattered electron detector. Images were acquired with the Zeiss SmartSEM V05.06 imaging software.

#### 2.2.7. Stability in Simulated Gastric and Intestinal Fluid

To estimate the stability of the archaeosomes in the microenvironment of the stomach and intestine, the following simulated fluids were used. Gastric simulated fluid (GSF) was prepared using sodium chloride, pepsin, and hydrochloric acid with a pH adjusted to 1.2. For the simulated intestinal fluid (SIF) with a pH of 6.8, monobasic potassium phosphate, pancreatin, and sodium hydroxide were used [25]. The solutions were prepared according to the U.S. Pharmacopeia. Then, 1 mL of archaeosome solution (7 mg/mL) loaded with calcein dye was mixed with simulated body fluids in a volume ratio of 2:1 and inserted in a dialysis tube with a 100 kDa molecular cut-off membrane. The filled tube was immersed in 15 mL PBS buffer and continuously stirred using a magnetic stirrer. The experiments were performed at room temperature. Every hour for the first 6 h and after 24 h a 100 µL sample was taken from the surrounding PBS buffer. The displaced volume was refilled using the same amount of PBS buffer. After 24 h, a disruption of the archaeosomes was induced by adding 10 µL of 10% Triton X-100 to release calcein quantitatively. The fluorescence intensities were measured in a 96-well plate with a CLARIOstar plate reader (BMG LABTECH GmbH, Ortenberg, Germany) at excitation and emission wavelengths of 475 nm and 530 nm, respectively.

#### 2.2.8. In Vitro Uptake Studies

To simulate the intestinal barrier a Caco-2 and HT29-MTX co-culture model was used. Caco-2 cells are epithelial-like cells isolated from a patient with colorectal carcinoma, while the HT29-MTX are differentiated into mature goblet cells with primary mucus-secreting function. Cell lines were combined at a ratio of 7 to 3 of Caco-2 to HT29-MTX and seeded in a total amount of 500,000 cells/well onto the translucent membrane of a Thincert cell culture insert of a 12 trans-well plate system (Greiner Bio-One, Kremsmünster, Austria). Further cultivation was necessary until reaching the full confluency of the monolayer. To determine the integrity of the cellular barrier the transepithelial electrical resistance (TEER of 427 ± 18 Ω/cm^2^; n = 12) was measured using the Millicell ERS-2 system (Millipore Corp., Burlington, MA, USA) [30]. Following this, samples were loaded on top of the cell layer within the apical compartment and incubated at 37 °C for 4 h, resembling the typical environment and duration of digestion. The basal compartment was filled with 1 mL high-glucose DMEM solution without phenol red. Three different archaeosome formulations were used. The first formulation was loaded with calcein, the second was labeled with 0.003 mol% fluorescent DOPE-rhodamine lipid, and the third formulation contained both calcein and the fluorescent DOPE-rhodamine lipid within the lipid layer. All formulations were diluted to a final concentration of 500 µg/mL lipid. As a positive control a 0.05 mg/mL calcein solution was used, while the negative control consisted of only PBS buffer [31]. After a 4 h incubation period the contents of both compartments were collected and calcein-labeled (green) and rhodamine-labeled (red) archaeosomes were quantified by fluorescence assays at wavelength pairs of excitation/emission of 475/530 nm and 560/597 nm, respectively. This approach enabled a detailed examination of the efficacy of archaeosome formulations to cross the cellular barrier or to adhere to and penetrate the cell layer, allowing for the separate analysis of lipid components and entrapped drugs (calcein).

Within the same set of experiments, the cell monolayer was fixed onto the trans-well membrane using 10% formaldehyde. Subsequently, the cells were stained with Hoechst 33,258 (blue) and two sets of phalloidin dyes: Phalloidin Rhodamine 565 (red) and Phalloidin Alexa Fluor 488 (green) depending on the specific formulation used. After 30 min of incubation the cells were washed twice with buffer. Visualization of the prepared and stained specimen was performed using a Zeiss LSM 510 confocal laser scanning microscope (Carl Zeiss AG, Oberkochen, Germany). For every randomly chosen area a stack of approximately 20 images was captured, changing the focal height systematically and thus recording variations in the color intensities along the *z*-axis, covering a distance of about 10 µm. From these data, a 3D depth profile was created [32].

#### 2.2.9. Lyophilization

Archaeosomes loaded with insulin were transformed into a solid dosage form. The lipid concentration was maintained at 7 mg/mL and the insulin concentration was about 3 mg/mL. The formulation was enriched with lactose at a final amount of 5% (*w*/*w*). Lactose served as a cryoprotectant during the drying process forming a matrix in which the archaeosomes are incorporated. Aliquots of 5 mL of archaeosomes were transferred into 50 mL lyophilization glass vials, which were then frozen overnight at −80 °C. The lyophilization step was carried out using the Benchtop 3L Sentry device (VirTis, Gardiner, NY, USA). The device operates under reduced pressure causing the sublimation of frozen water, leaving behind a dry powder containing the archaeosomes. The powder particles were then characterized using SEM. After the reconstitution of the dried particles in purified water SEM, PCS, native gel electrophoresis, and a BCA protein assay were performed.

#### 2.2.10. Spray Drying

The same formulations as those for the lyophilization process along with 5% (*w*/*w*) lactose were employed for spray drying. The Büchi B-90 Nano Spray Dryer (BÜCHI Labortechnik AG, Flawil, Switzerland) was used. The drying parameters were kept constant at an inlet temperature of 120 °C and an air flow of 120 L/min, resulting in a pressure of 45 mbar(g). To enable a stationary state, the whole system was conditioned by spraying the pure solvent (i.e., water) before each sample run for a minimum of 20 min. This ensured a constant temperature of the spraying head of approximately 84 °C. The spray rate was set at 30%, generating a fine constant cone of spraying mist. Being dispersed in hot air, the droplets evaporated, leaving behind fine powdered particles that adhered to the cylindrical collecting electrode situated at the bottom of the device. The analysis of the powdered particles and rehydrated archaeosomes was performed in the same way as after the lyophilization process.

#### 2.2.11. Statistical Analysis

Results presented are expressed as mean values ± standard deviation (SD). If not stated otherwise, n = 3. For statistical analysis of the data, a Student’s *t*-test was performed, and differences were considered significant at levels of *p* ≤ 0.05.

## 3. Results

### 3.1. Manufacturing and Physicochemical Characterization of Archaeosomes

Efficient production of archaeosomes, derived from diether and tetraether lipids isolated from *Sulfolobus acidocaldarius*, was achieved using microfluidic mixing methods after carefully adjusting the process parameters, including the total flow rate (TFR) and the flow rate ratio (FRR) between the aqueous and the organic phase. Constrained by the solubility of the archaeal lipids, the highest achievable lipid concentration was 20–21 mg/mL in both DMSO-iPA and absolute ethanol, which were used as organic phase. To establish the microfluidics mixing protocol we first varied the FRR of aqueous to organic phase from 2:1 to 5:1 and 10:1. For any of these conditions, TFR was varied from 1, 2, 5, 10, to 12 mL/min. Increasing the FRR in favor of water resulted in a decrease in particle size to 10–20 nm for a ratio of 10:1 aqueous to organic phase. These archaeosomes showed a low encapsulation efficiency and were unstable in solution regardless of the TFR. Thus, an FRR of 2:1 was used for the following experiments, again varying the TFR from 1 to 12 mL/min as described above. It was found that with increasing TFR the particle size decreases while the PDI value tends to increase. Using a TFR of 1 and 2 mL/min, the particle sizes were below 200 nm. Thus, to attain a particle size of about 100 nm the TFR was adjusted to 1.7 mL/min. The FRR was set and kept constant at 2:1 in favor of the aqueous phase, but still resulted in high amounts (about 30%) of organic solvents within the formulation. Immediate dialysis against HEPES buffer effectively reduced this solvent content. The size of the empty archaeosomes determined after dialysis by PCS was 85.4 ± 0.6 nm with a PDI value of 0.34 ± 0.02. The archaeosomes showed a highly negative zeta potential (**ζ** = −31.8 ± 0.2 mV), indicative of a pronounced physical particle stability. This premise was also supported by the long-term stability experiments, which were conducted over a period of 6 months. During storage at 4 °C the average size of the archaeosomes changed only slightly to 93.22 ± 2.7 nm, with a PDI of 0.42 ± 0.24, consistent with an unchanged transparency of the sample. A summary of all size data is provided in Table 1. It is important to note that for the stability assays, cellular uptake studies, and powder production, different batches of natural lipid extracts have been used, nevertheless, the process parameters remained the same. The data suggest that archaeosomes can be efficiently produced through microfluidics when employing optimized operational parameters [33]. Accordingly, using a continuous flow pump system in conjunction with an array of mixing channels would enable the continuous large-scale manufacturing of archaeosomes relevant for clinical applications [34,35]. The size data were affirmed by TEM measurements of negatively stained air-dried archaeosomes. The TEM images showed single particles in a proper size range around 100 nm (Figure 2a). By SAXS measurements, information on the internal structure, lamellarity, and lipid layer thickness of archaeosomes could be derived. The SAXS scattering profile (Figure 2b) showed the characteristics of an uncorrelated layer structure typical for unilamellar vesicles. Thus, the data were fitted to a unilamellar vesicle model by using a single bilayer electron density model. The thickness of the archaeal lipid bilayer was derived from the positions of the maxima in the electron density profile (Figure 2b, inset). Defined as lipid headgroup-to-headgroup distance an average value of ~4.5 nm was obtained for the membrane thickness of the archaeosomes. This value is in good agreement with published data, which revealed a thickness of the central hydrophobic part of 3.0 nm plus an outer hydrophilic shell thickness of ~1.5 nm for archaeal lipid layers [36]. Compared to liposomes composed of conventional phosphatidylcholines whose bilayer thickness is in the range of 3.0–4.1 nm depending on lipid composition, temperature, and salt content [37], archaeosomal membranes are substantially thicker.

### 3.2. Archaeosome Stability in Simulated Intestinal Fluids and Drug Release Behavior

Stability, leakage, and in vitro release kinetics were probed using calcein as a model drug. The calcein molecules were incorporated into the archaeosomes directly during the manufacturing process. Calcein loading induced a noticeable modification of the particle size distribution, resulting in larger but highly uniformly sized particles with a very low PDI value (Table 1). One possible explanation for this effect could be the high calcein concentration, which increases the ionic strength of the solution, potentially enhancing the electrostatic repulsion between the single particles or even changing the curvature of the lipid bilayer. Notably, the particle size and homogeneity remained unchanged after dialysis and removal of the organic solvent and non-encapsulated calcein. To study the permeability of the lipid membrane and particle stability in different media, we monitored the fluorescence signal intensity of calcein. At the initial relatively high concentration of calcein encapsulated within the archaeosomes, the intrinsic fluorescent signal of calcein is self-quenched and the fluorescence intensity is low. Upon leakage and release of calcein from the archaeosomes to the surrounding aqueous environment, the fluorophore becomes diluted and the fluorescence intensity increases in a concentration-dependent manner. This behavior allowed us to follow the release profile of calcein under different environmental conditions. In particular, we investigated the release properties of archaeosomes in PBS buffer (pH 7.4), simulated intestinal fluid (SIF) (pH 6.8), and simulated gastric fluid (SGF) (pH 1.2), with and without the addition of the characteristic digestive enzymes pancreatin and pepsin, respectively. A modified protocol for the in vitro digestion of food in the gastric and intestinal phase, which recommends an incubation for 2 h at 37 °C, was applied [38]. As we did not see any release after 2 h, we repeated the whole experiment for 24 h but this time at 20 °C to follow particle stability over time rather than digestion. This approach allowed us to better understand the behavior of archaeosomes under different conditions and time scales. Across all samples, a minor release was observed within the first few hours, which was minimal for samples incubated at pH 7.4 in buffer and at pH 6.8 in SIF, regardless of the presence of pancreatin. At pH 1.2 the overall release rate was slightly higher but still remained below 3%, irrespective of the presence of pepsin. After a certain time, the release curves flattened out, indicating no further release of calcein from the particles (Figure 3). The very low release rate observed over 24 h demonstrated the high stability of the archaeosomes, even when exposed to extreme pH conditions. Just after chemical destruction of the lipid membrane with solubilizing detergents (addition of 10 µL, 10% Triton X-100) an immediate and rapid release of the payload could be induced, considered as 100% release. For oral drug delivery the elevated stability implies that the particles readily survive the harsh conditions of the stomach (simulated by SGF), however, the particles are equally stable in the intestinal environment (simulated by SIF). Additionally, degrading enzymes had no marked impact on the membrane stability of the archaeosomes as observed in vitro using a commercially available porcine pancreatin extract. This extract is commonly used and contains a mixture of proteases, amylases, and lipases with well-characterized enzyme activities. However, the phospholipase activity in the porcine pancreatin extract is four times lower than that found in human pancreatic juices, where degradation is more likely to occur [39]. Nevertheless, our data support the general consensus that archaeosomes are less susceptible to enzymatic digestion by lipases when compared to conventional liposomes composed of ester lipids [12,40].

### 3.3. In Vitro Release and Cellular Uptake Behavior

For the in vitro release and uptake studies a co-culture cell system mimicking the intestinal epithelium in combination with fluorescence spectroscopy and confocal laser scanning microscopy (CLSM) was used. For the fluorescent uptake assay, freshly prepared formulations were pipetted on an already confluent co-culture (Caco-2 and HT29MTX) cell layer within the trans-well system. After 4 h of incubation, resembling the duration of digestion, fluids were collected from both the apical and the basal compartment. Quantification of archaeosomes was made possible by the use of rhodamine-labeled lipids. The fluorescence intensities were measured at the beginning for all formulations which had been pipetted on the cells (insert) and after incubation from the apical and basal compartments. The intensity measured for the formulation initially placed in the trans-well system was defined as 100% of particles available for the interaction with the cell layer. The percentage from the apical compartment corresponded to the portion of recovered particles not interacting with the cells, while the percentage of particles in the basal compartment represented the amount that went through the cell layer and the polycarbonate membrane, corresponding to the archaeosomes that passed through the model membrane of the intestinal barrier. The portion of archaeosomes that were either attached to or taken up by the cells was calculated by subtracting the percentages in both apical and basal compartments from the initial total amount determined in the insert. The results (Table 2) revealed that approximately 27–29% of both calcein-loaded (TR_C) and empty (TR) archaeosomes were associated with the cells. The percentage of the archaeosomes found in the basal compartment was negligible. Taken together these findings suggest that intact archaeosomes cannot pass through the cell layer, but a substantial proportion (about 30%) of the archaeosomes is associated with the cells.

To determine the in vitro release kinetics of encapsulated calcein, the same experimental setup was used as described above. After an incubation time of 4 h the fluids were collected in the apical and basal compartments and the concentration of calcein was determined before and after addition of Triton X-100 (10 µL of 10% (*v*/*v*) solution). The addition of detergent was required to quantitatively release calcein, thereby preventing signal quenching of calcein molecules entrapped within archaeosomes. Similar to the assay described above, the fluorescence intensity of the initially loaded calcein in the insert was defined as 100%. The percentages of calcein present in the apical and basal compartments were determined after collection by relating the measured fluorescence intensities to the initial fluorescence intensity. The results are presented in Table 3.

The amount of calcein released from archaeosomes was relatively low, ranging from 1–3%. This finding is roughly consistent with the release behavior of calcein from archaeosomes incubated with simulated body fluids. However, when the formulations were additionally enriched with the fluorescently labeled lipid DOPE-Rhodamine, the release rate of encapsulated calcein increased to approximately 16–19%. These data suggest that the addition of conventional lipids, even at low concentrations, has an impact on the drug release properties of archaeosomes. Given that the DOPE phospholipid (0.003 mol%) was incorporated in the lipid membrane, it is tempting to speculate that the DOPE molecules, which contain two unsaturated fatty acid chains with cis double bonds that cause a kink in the hydrocarbon chain, introduce some disorder in the archaeal membrane. Defects in the lipid membrane could make it more permeable, thereby facilitating calcein release.

To study the localization of archaeosomes with respect to the cells more closely we employed confocal laser scanning microscopy (CLSM). Three different kinds of formulations containing differently colored fluorescent labels were prepared for double labeling of archaeosomes, alongside with blanks (PBS) and calcein solution serving as controls. The double labeling approach allowed the simultaneous visualization of calcein along with archaeosomes whose lipid membrane was labeled with a fluorescent DOPE lipid component. Focusing at varying depths along the *z*-axis allowed the visualization of distinct layers within the cell monolayer, as shown in Figure 4a–f. The cytoskeleton was stained with two phalloidin stains, either in red or in green, and served as control to provide information about the cell layer thickness. As seen (Figure 4a,b), the cells formed a continuous monolayer as described previously [30]. The differently colored cytoskeleton (green or red) enabled the distinction between the positions of archaeosomes (red) and calcein (green) in relation to the cytoskeleton. The blue color originated from the Hoechst stain bound to DNA, staining the nuclei of the cells. A free calcein solution was applied as control (Figure 4c). When calcein was loaded into archaeosomes, the green color was detected on top of the red stained layer of the cytoskeleton (Figure 4d) with very low penetration of calcein into the cell layer. Combining this information with the data from fluorescence spectroscopy (Table 3), it can be stated that the majority of the loaded calcein remained entrapped inside the archaeosomes. This finding is further confirmed by the co-localization (yellow) of archaeosomes (red) and calcein (green) as seen in Figure 4f. Here the cell membrane was not stained, but the nuclei seen in blue allowed an estimation of the spatial position of archaeosomes and calcein with respect to the cell membrane. When empty archaeosomes labeled with the fluorescent lipid DOPE-Rhodamine were used (Figure 4e), a co-localization (yellow) with the cytoskeleton is observed, indicating that the archaeosomes reach the upper part of the cells. These results align with the analytical data presented in Table 2, where almost 30% of the archaeosomes adhered to the cell membrane. The co-localization of red and green indicated that most calcein was still entrapped inside the archaeosomes, while empty archaeosomes after calcein release are visualized in red (Figure 4f). The results of the uptake images correlate nicely with data from fluorescence spectroscopy presented within Table 2 and Table 3.

### 3.4. Dry Powder Formulations of Archaeosomes

To investigate the applicability of archaeosomes as dry powder formulations, both empty or insulin-loaded particles were either spray-dried or lyophilized in the presence of 5% lactose as cryoprotectant. The corresponding SEM images of the powdered formulations embedded in the sugar matrix are presented in Figure 5. The SEM images of the spray-dried archaeosomes showed differently sized single spherical particles (Figure 5a). Notably, when insulin was encapsulated in the spray-dried archaeosomes, the images revealed the presence of larger spherical particles exhibiting a distinctive crinkled pattern (Figure 5b). This unique appearance was not observed in the case of empty archaeosomes or those filled with calcein. Despite the observation of various spray-dried nanoparticles using lactose as sugar matrix, the underlying causes of the different morphologies observed for the spray-dried material remain unclear. One possible explanation for wrinkled surfaces may be attributed to temperature effects like rapid shrinkage of the powder during cooling [41]. Additionally, the interaction with the wall material has been suggested as another potential reason for diverse surface structures [42]. In contrast, the lyophilized powder showed a spongy bone-like structure that looked very fragile. The fragility was even more pronounced in the presence of insulin (Figure 5c,d). The dried powder formulations were not further processed except for rehydration with water. Figure 6 shows the SEM images of the rehydrated insulin-loaded archaeosomes, where individual particles of appropriate size are distinctly recognizable.

### 3.5. Insulin Loading for Dry Powder Formulations

Insulin was used as a model peptide drug and encapsulated into archaeosomes in the course of the microfluidics-assisted self-assembling process of archaeosomes. The insulin-loaded archaeosomes exhibited an average size of 97.5 nm and a PDI value of 0.285, comparable to empty ones. The samples were then lyophilized and spray dried as described above. The powder formulation was subsequently redissolved with pure water for size measurements. Both the mean particle size and the PDI values slightly increased after rehydration to 105.9 nm and 103.1 nm with PDI values of 0.348 and 0.324 for spray-dried (SD) and lyophilized (LYO) samples (Table 4). It is notable to mention that the archaeosomes have to endure extreme temperature ranges throughout these specific drying processes. For lyophilization, the sample needed to be frozen at −80 °C, while the spray-drying process was performed at elevated temperatures (inlet air 140 °C, head ~80 °C, bottom ~50 °C). Although the particles do not experience the maximum temperature due to the cooling effect of the evaporated solvent, the archaeosomes will experience heating in the laminar flow of the drying gas. Throughout both processes the archaeosomes have demonstrated exceptional stability across all temperature ranges, remaining largely unaffected by the conditions inherent in these processes at least in their size distribution.

To determine the encapsulation efficiency (% EE) of insulin, the concentration of insulin was determined using the BCA protein assay. The % EE was calculated as
% EE = (*c_total_*/*c_inc_*) × 100%(1)
where *c_total_* is the initial amount of insulin used and *c_inc_* is the insulin concentration found after microfluidics preparation and removal of unencapsulated insulin. To determine *c_inc_* the archaeosomes were disrupted with Triton X-100 to quantitatively release the entrapped insulin. After preparation, about 35% of the initially used insulin was encapsulated (Table 4). After the drying and rehydration process, the overall recovery of insulin decreased by approximately one-third. It can be assumed that during drying procedures and handling of the powdered material a substantial amount of archaeosomes has been lost. As the protein assay cannot distinguish between archaeosomal entrapped insulin and released free insulin in solution, native gel electrophoresis was performed.

This method not only allowed us to distinguish between the localization of insulin (entrapped or free) but also exhibited a much higher sensitivity compared to conventional protein assays. As seen on the native gels (Figure 7), the insulin standard migrated a defined distance into the gel according to its molecular weight and charge (lane 2). For archaeosomes containing encapsulated insulin no discernible band of free insulin was visible (lane 3), unlike to archaeosomes that were disrupted with detergent prior to gel loading (lane 4), thus confirming successful loading. Comparable results were obtained for the redissolved powdered formulations, where no noticeable release of insulin was detected in the gel neither for the spray-dried (lane 5) nor for the lyophilized (lane 7) formulations. However, upon disruption of the archaeosomes by addition of detergent, a single band corresponding to insulin became evident (lanes 6 and 8) indicating the release of insulin. Notably, no indications of insulin oligomerization or aggregates became visible in the gel. This analysis confirms the successful encapsulation and retention of insulin within the archaeosome formulations throughout the drying processes.

The primary aim of creating powdered formulations was to confirm the stability of archaeosomes during the drying process without releasing their payload. Based on the promising findings we propose drug-loaded archaeosomes as good candidates for oral delivery in tablet form. Accordingly, the dry powders containing insulin-loaded archaeosomes and a lactose matrix as a filler could be transformed into an oral tablet by adding a binder or proceeding with dry granulation by applying pressure to form a pill. Considerations regarding the final dose to be delivered are challenging due to its dependency on various factors including the class of the active ingredient and individual patient-specific needs. In particular, for the delivery of insulin parameters like body weight, insulin sensitivity, and current blood glucose level have to be considered. In this context, a slow-release formulation could offer significant advantages by providing a sustained and constant supply of insulin over a certain time.

## 4. Discussion

Efficient production of archaeosomes, derived from diether and tetraether lipids isolated from *Sulfolobus acidocaldarius*, was achieved using a microfluidics-based mixing method [33,43]. While batch process synthesis remains the standard in the pharmaceutical industry, flow-based manufacturing processes have emerged over the past decade. With this simple self-assembling synthesis protocol, archaeosomes could be fabricated and loaded with drugs in a single step. This is an important prerequisite for the establishment of a continuous manufacturing process in compliance with good manufacturing practice (GMP) standards, which are a critical requirement for the pharmaceutical industry. Certain parameters must be considered when a microfluidics-assisted manufacturing process is used for the production and simultaneous loading of drugs in archaeosomes. These parameters are generally the same as those for the microfluidic production of liposomes [31]. They include the type and concentration of lipids, kind of organic solvent, and the physicochemical properties of the compounds to be encapsulated. Among these, the key factors influencing the manufacturing process and the characteristics of the final product are the total flow rate (TFR) and the flow rate ratio (FRR) between the aqueous and the organic phases. However, it is equally important to consider the relationship between those parameters, particularly assessing the compatibility between substances dissolved either in the organic or aqueous phase. Keeping in mind that the archaeosomes are intended for drug delivery, the primary objective was to keep the lipid concentration as high as possible to enhance the encapsulation efficiency while minimizing the concentration of the organic solvent within the formulation. The latter point was highly important as an exposure to high concentrations of organic solvents resulted in particle destabilization and aggregation. It is needless to stress that such conditions could also negatively impact the encapsulated compounds (e.g., causing protein aggregation or rapid drug release). Considering all these aspects, the values for TFR, FRR between aqueous and organic phase, lipid concentration, buffer system, and notably salt content were systematically adjusted. However, it has to be mentioned that depending on the individual batches of TEL, the choice of the organic solvent, and the loaded compounds, some variations in the particle size distribution and the PDI values were observed, as shown in Table 1. In this way, archaeosomes of tailored particle sizes exhibiting good encapsulation efficiencies were produced while maintaining remarkable stability over 6 months when stored at 4 °C. Using the hydrophilic model compound insulin, a successful encapsulation within the aqueous core of archaeosomes could be demonstrated in a continuous flow process. Encapsulation of hydrophilic compounds in the absence of any driving force that will stimulate loading efficiency leads to an even distribution of the active ingredient in the inner particle volume and the total liquid preparation volume. One strategy to increase encapsulation efficiency would be to increase the particle size, occupying more space within the same liquid volume. Another option would be to increase the particle concentration in the formulation. Increasing particle concentration necessitates higher lipid solubility within the organic phase, which could be achieved by an exchange of the organic phase. Consequently, improved solubility of tetraether lipids in the organic phase would allow employment of higher flow rate ratios in favor of the aqueous phase. This, in turn, would lead to a lower concentration of organic solvent and an increased concentration of archaeosomes in the final product. Taking all these considerations into account, the maximal solubility of the archaeosomal lipids in both ethanol and DMSO/propanol was limited to approximately 20–21 mg/mL. The optimal process parameters to obtain homogeneous archaeosomes of about 100 nm in size were TFR 1.7 mL/min with an FFR of 2:1 aqueous to organic phase. Despite thorough dialysis against the buffer, it was estimated that about 1–2% of organic solvent remained in the samples. For the experiments presented, this concentration was entirely acceptable. However, for continuous flow production of larger volumes, a tangential flow filtration system could be connected to remove the organic solvent quantitatively.

A detailed study on the stability of archaeosomes in different media has shown an ideal behavior in simulated gastric fluid, suggesting that archaeosomes can easily withstand the stomach content and safely transport the encapsulated drug to the intestine. The pronounced stability of archaeosomes was also retained in simulated intestinal fluid, even in the presence of phospholipases that typically degrade membranes [12]. Given that the target site of the archaeosomes is the small intestine, this finding poses a challenge when the drug is to be released on-site. To address this issue, one option would be to destabilize the structure of the lipid membrane by increasing membrane fluidity and permeability. This could either be achieved by direct modifications of the archaeal lipids, for example, by modifying their typical growth conditions [44] or by alterations of the lipid composition in the archaeosomes e.g., by addition of conventional unsaturated lipids or fatty acids, which are known to have destabilizing effects on membranes [45]. Indeed, even a slight change in the lipid bilayer composition by addition of conventional phospholipids (0.003 mol % DOPE) resulted in a higher drug release rate. Accordingly, most studies published so far use hybrid/mixed vesicular systems or conventional liposomes that are enriched with archaeal lipids for drug delivery [13,46,47].

To allow efficient uptake by intestinal epithelial cells, the archaeosomes need to have a certain size. If they are too large they cannot be taken up by cells and, even more importantly, the particles may not diffuse fast enough into the mucus layer [48]. While the archaeosomes synthesized in this study had an appropriate size, they did not cross the intestinal barrier after 4 h of exposure as evident in the in vitro uptake assay. This could be due to the presence of the mucus layer present in the Caco-2/HT29-MTX culture system. The mucus mesh has hydrophobic regions hampering the free passage of particles that have to penetrate the mucus layer at rates faster than mucus renewal and clearance, which is estimated to be 50–270 min [49,50]. Nevertheless, the archaeosomes revealed a remarkable affinity (about 30%) to the cell membrane. This enhanced adhesion affinity is expected to prolong the timeframe for retarded release of encapsulated drugs, facilitating the absorption through the intestine wall.

So far, first studies have indicated the potential utility of archaeosomes for oral drug delivery, showing corresponding in vivo results [12,13]. For example, an in vivo study using insulin-loaded hybrid archaeosomes composed of a mixture of archaeal lipids and synthetic phospholipids orally applied in diabetic mice revealed that insulin was retained significantly longer in the GI tract when stabilized by archaeosomes. Significant lower levels of blood glucose were found when compared to conventional liposomal formulations [14]. However, similar to our studies, the in vitro transport across Caco-2 cell monolayers was limited. The pronounced stability of archaeosomes is probably due to the unique structure of the lipids, the tight membrane organization, and the negatively charged surface of the archaeosomes [51,52]. These distinct structural characteristics apparently improve the stability of proteins, peptides, and water-insoluble compounds in the GI tract, whereas archaeosomes are not readily taken up in the intestine [15,53]. Parmentier et al. observed the presence of radiolabeled ether lipids in the inner organs of rats and subsequently in the urine after renal excretion following oral administration of liposomes containing specific mixtures of conventional phospholipids and ether lipids. From this, the authors proposed that the hybrid liposomes adsorb to enterocytes, enabling the drug to permeate into the bloodstream after being released, while only a negligible fraction of the hybrid liposomes were taken up [54]. In contrast, Morilla et al. detected a considerable amount of a radiolabeled substance with low intestinal permeability in the blood of rats after oral administration of archaeosomes composed exclusively of archaeols. The authors attributed this finding to the uptake of intact archaeosomes by M cells in Peyer’s patches, while drug leakage occurs during transcytosis [55]. Within this context, various types of archaeosomes characterized by different molecular lipid compositions and structural features have been proposed as oral vaccines targeting M cells within Peyer’s patches. There, archaeal glycerolipids act as strong adjuvants mediating T-cell immune responses [56,57]. These studies have demonstrated that an enhanced in vivo uptake of archaeosomes by M cells is linked to an increased surface hydrophobicity of the archaeosomes and larger particle sizes exceeding 400 nm [55]. However, the limited number of M cells in the intestine has hindered the effective targeting of M cells and subsequent antigen delivery into mucosa-associated lymphoid tissue. Hence, it is more likely that the archaeosomes interact with goblet cells and the mucus layer that covers the intestinal epithelium. To improve mucus penetration the hydrophobic and electrostatic interactions with the mucus layer may be reduced by modifying the surface properties of the archaeosomes. Another strategy to enhance mucus penetration and cellular uptake could be to modify the surface of archaeosomes with cell-specific targeting ligands. Cell surface binding might be an important step for subsequent internalization of the particles via various non-specific endocytosis pathways for intracellular drug release. Alternatively, upon mucus layer adhesion drug absorption could occur via an adsorption mechanism and/or through passive permeation of the released drug. Thus, the underlying intestinal uptake mechanisms of archaeosomes remain uncertain.

Notably, current literature has not reported any adverse side effects associated with orally administered archaeosomes. In fact, the in vivo studies rather emphasize the remarkable biocompatibility, low toxicity, and the good safety profile of archaeosome formulations [46]. These findings underline the high potential of archaeosomes as safe and effective delivery systems for various applications.

In this study, insulin, a small peptide hormone, was chosen as a model drug for various reasons. First of all, the number of diabetic patients depending on a constant exogenous supply of insulin is steadily increasing and diabetes has emerged as one of the most prevalent non-communicable diseases worldwide [58]. This makes insulin one of the most commonly used peptide therapeutics. Today, insulin is typically administered by subcutaneous injections, which are often associated with injection pain, patient discomfort, and local infections [59]. Consequently, extensive efforts have been made to explore non-invasive methods for insulin administration to improve patient compliance. Among these, the oral route is the most patient-friendly and convenient way for drug administration. However, challenges that impede effective oral delivery of insulin include rapid inactivation by proteolytic enzymes in the GIT and low epithelial permeability due to its large molecular weight, hydrophilicity, and negative net charge at neutral pH [60]. Moreover, the transport of insulin across the mucus layer, which functions as the initial diffusional and enzymatic barrier to intestinal uptake, is probably hindered by electrostatic repulsion due to the negatively charged mucus layer [61]. Thus, to improve intestinal uptake and bioavailability of insulin, various carrier systems have been developed to shield insulin from enzymatic degradation and enhance its intestinal transport [62]. So far, no ideal oral delivery system has been reported to transport insulin at an acceptable level to circulation. While oral delivery of insulin remains an unsolved issue, the field of oral peptide delivery is rapidly emerging as biological macromolecules gain increasing significance. In this regard, lipid-based delivery systems including archaeosomes offer a promising platform for improving the efficacy of therapeutic peptides [63]. Moreover, it is noteworthy to highlight that archaeosomes can serve as delivery vehicles not only for peptides but also for small molecules, oligonucleotides, proteins, or antibodies.

Here, it was shown that archaeosomes can be easily fabricated and loaded with insulin using microfluidics technology, enabling continuous flow large-scale production in the future. The encapsulation efficiency of insulin was satisfying and the archaeosomes could be lyophilized and spray dried without any detectable release of their payload. These findings underscore the pronounced resilience of insulin-loaded archaeosomes, making them a viable option for oral medication as tablets.

## 5. Conclusions

This study aimed to provide a better understanding of archaeosome-based drug delivery systems from their manufacturing process and physicochemical characterization to their interaction with the intestinal barrier and, finally, the development of a stable solid dosage form. While insulin serves as a model drug, archaeosomes will function as a versatile delivery system for a wide range of pharmaceuticals. However, adapting process parameters and conducting physicochemical tests remain necessary when encapsulating other pharmaceutical substances. Despite this requirement, the selected methodologies, specifically single-step microfluidics-assisted synthesis with constant drug loading alongside the production of dry powder formulations as presented in this paper, effectively address some challenges associated with the formulation and manufacturing process of peptide drugs for oral administration.

## Figures and Tables

**Figure 1 pharmaceutics-16-00694-f001:**
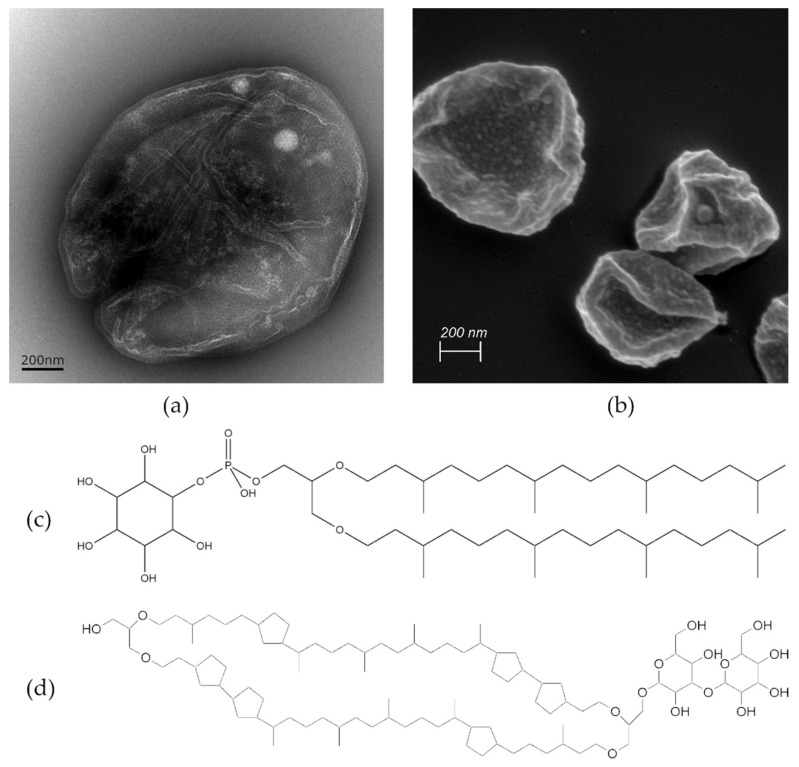
Visualization of *Sulfolobus acidocaldarius* cells using (**a**) TEM and (**b**) SEM. The scale bar represents 200 nm. The sample preparation for SEM includes water displacement through a series of ethanol treatments that can cause cell shrinkage as evident in (**b**). The molecular structures of two ether lipid species isolated from the archaeal membrane are presented in (**c**,**d**), highlighting structural differences. (**c**) Features archaeol—the principal representative of DEL containing two isoprenoid chains and (**d**) caldarchaeol—a prominent representative of TEL, which is almost double in length having two polar groups connected with two isoprene-like chains containing 6 cyclic structures.

**Figure 2 pharmaceutics-16-00694-f002:**
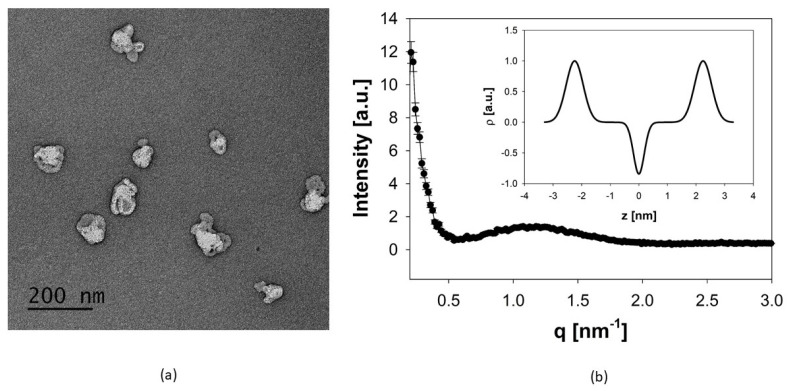
(**a**) TEM image of negatively stained archaeosomes using 1% uranyl acetate. The image of the air-dried archaeosomes shows single particles with fairly uniform size distribution. The scale bar represents 200 nm. (**b**) Global fit of the SAXS pattern of archaeosomes indicating unilamellar structures. The inset gives the corresponding electron density profile, in which the distance between the two maxima of the Gaussian modeling of the electron dense headgroup regions gives a measure for the lipid layer thickness.

**Figure 3 pharmaceutics-16-00694-f003:**
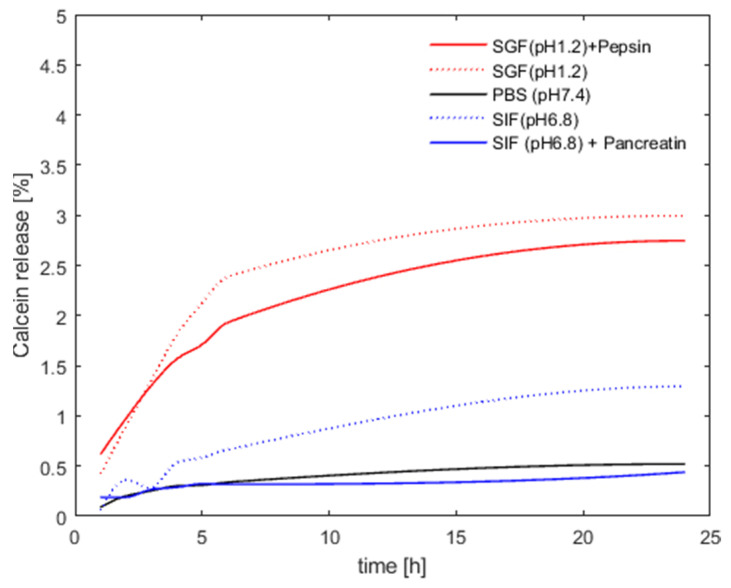
Archaeosome stability in different media. The presented values are given as percentages of released calcein compared to the maximum fluorescence signal obtained after detergent-induced membrane disruption (defined as 100% release).

**Figure 4 pharmaceutics-16-00694-f004:**
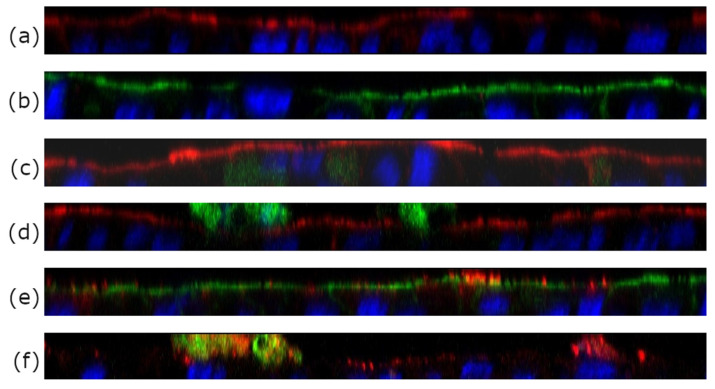
CLSM z-scans of co-culture (Caco-2/HT_29MTX of 7/3) cell layers incubated with different archaeosome formulations. (**a**,**b**) Blank samples incubated with pure buffer only. The cytoskeleton was labeled with phalloidin (either red or green), the nuclei were stained with Hoechst Blue. (**c**) Free calcein. (**d**) Calcein-loaded archaeosomes. The green stain detected on top of the red cytoskeleton layer implies that the calcein is found outside on top of the cell membrane with low penetration of calcein into the cell layer. (**e**) Co-localization (yellow) of red-labeled empty archaeosomes and the cell layer (in this case colored with a green phalloidin dye) indicates good adhesion of archaeosomes to the cell membrane. (**f**) shows the co-localization in yellow of calcein (green) still encapsulated within archaeosomes (red). In this case the cytoskeleton was not stained.

**Figure 5 pharmaceutics-16-00694-f005:**
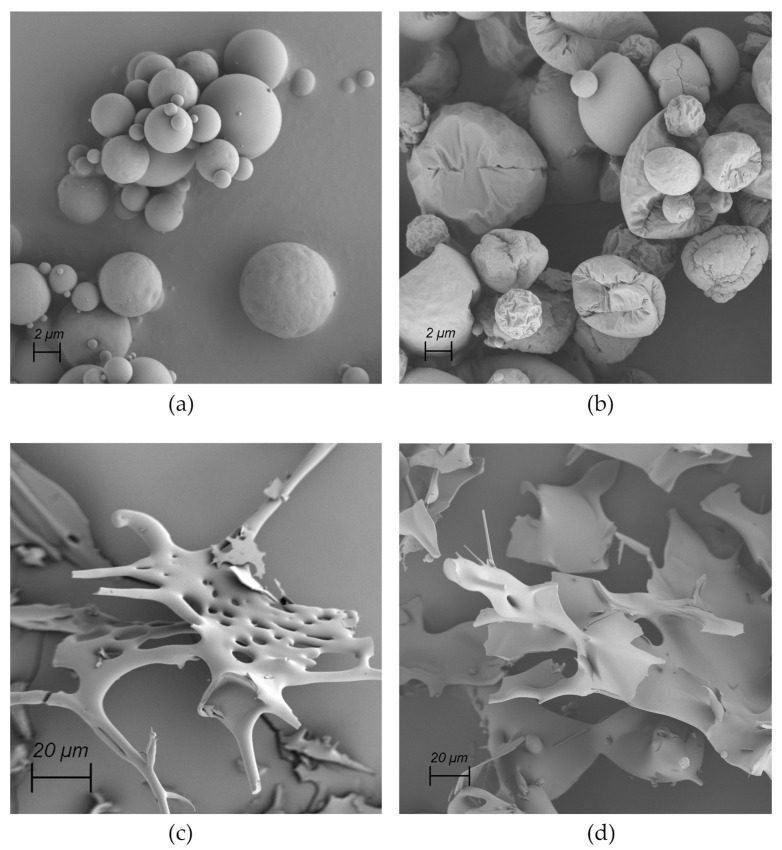
SEM images of dried powders. (**a**) Spray-dried empty archaeosomes. (**b**) Spray-dried archaeosomes loaded with insulin. (**c**) Lyophilized empty archaeosomes. (**d**) Lyophilized archaeosomes loaded with insulin.

**Figure 6 pharmaceutics-16-00694-f006:**
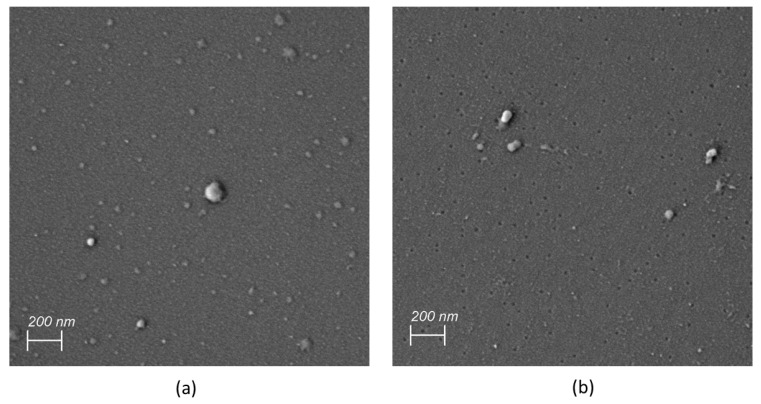
SEM images of redissolved insulin-loaded archaeosomes after spray drying (**a**) and lyophilization (**b**).

**Figure 7 pharmaceutics-16-00694-f007:**
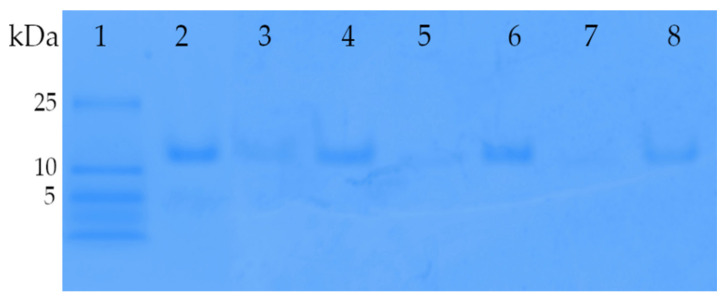
Acrylamide gradient native gel (4–12%) of insulin-loaded archaeosomes. Lane 1 presents the low-molecular-weight marker panel. Lane 2, reference insulin (1 µg) used for the encapsulation process. Lane 3, archaeosome formulation with encapsulated insulin directly after the preparation using microfluidics. The majority of insulin is encapsulated within the archaeosomes, resulting in minimal Coomassie blue staining. Only a faint band is visible due to a portion of non-encapsulated insulin. Lane 4 shows the same formulation mixed with Triton X-100, which disrupts the archaeosomes, leading to the release of encapsulated insulin. Thus, lane 4 corresponds to the amount of initially encapsulated insulin. Lanes 5 and 7 contain insulin-loaded archaeosomes after spray-drying and lyophilization processes, respectively. The powdered formulations were dissolved in water and loaded onto gel. The absence of insulin bands in lanes 5 and 7 suggests negligible insulin release during the drying processes. Lanes 6 and 8 display the same formulations after disruption with Triton X-100, confirming the presence of insulin inside the spray-dried and lyophilized archaeosomes, respectively.

**Table 1 pharmaceutics-16-00694-t001:** Overview of key experiments (stability assays, cellular uptake studies, powder production) with the respective archaeosome formulations used. All formulations were produced with the same process parameters, however, different batches of natural lipid extracts have been used. This might explain small deviations in size and polydispersity index (PDI) for identical formulations.

Organic Solvent	Loading	Mean Particle Size [nm]	PDI
Stability assays			
EtOH	-	85.4 ± 1.1	0.336 ± 0.017
DMSO-iPA (2:1 *v*/*v*)	-	86.3 ± 0.7	0.423 ± 0.140
DMSO-iPA (2:1 *v*/*v*)	calcein	111.6 ± 1.3	0.070 ± 0.002
Cellular uptake studies			
DMSO-iPA (2:1 *v*/*v*)	calcein	129.6 ± 0.7	0.119 ± 0.013
DMSO-iPA (2:1 *v*/*v*)	calcein + DOPE-rhodamine	127.0 ± 0.2	0.107 ± 0.006
DMSO-iPA (2:1 *v*/*v*)	DOPE-rhodamine	137.7 ± 0.5	0.220 ± 0.050
Powder production			
EtOH	insulin	97.5 ± 1.7	0.285 ± 0.031
EtOH	insulin *	103.1 ± 1.2	0.348 ± 0.079
EtOH	insulin **	105.9 ± 2.1	0.324 ± 0.064

* Spray-dried insulin formulation, re-dissolved in water.** Freeze-dried insulin formulation, re-dissolved in water.

**Table 2 pharmaceutics-16-00694-t002:** Quantification of archaeosomes labeled with DOPE-Rhodamine after the in vitro uptake assay using fluorescence spectroscopy (Ex 560 nm/Em 597.5 nm; n = 3). TR (archaeosomes labeled with DOPE-Rhodamine); TR_C (archaeosomes labeled with DOPE-Rhodamine and loaded with calcein).

	Inserted	Apical	Basal	Cell Attached
TR [%]	100	72.982 ± 1.672	0.063 ± 0.007	~27
TR_C [%]	100	70.994 ± 13.036	0.065 ± 0.009	~29

**Table 3 pharmaceutics-16-00694-t003:** Quantification of encapsulated and released calcein after the in vitro uptake assay using fluorescence spectroscopy (Ex 475 nm/Em 530 nm; n = 3). T_C (archaeosome loaded with calcein); TR_C (archaeosomes labeled with DOPE-Rhodamine and loaded with calcein).

		Inserted	Apical	Basal	Cell Attached
T_C	Calcein [%]	100	96.510 ± 0.382	0.418 ± 0.002	~3
TR_C	Calcein [%]	100	80.417 ± 0.385	0.443 ± 0.002	~19

**Table 4 pharmaceutics-16-00694-t004:** Mean particle size and polydispersity index (PDI) of insulin-loaded archaeosomes subsequent to preparation (T_ins), after spray drying (SD) and lyophilization (LYO), respectively (T_ins SD, T_ins LYO). Initial encapsulation efficiency % EE and overall recovery % after rehydration are shown (n = 3).

	Insulin	T_ins	T_ins SD *	TEL_ins LYO *
Mean particle size [nm]		97.5 ± 0.5	103.1 ± 1.2	105.9 ± 1.4
PDI		0.285 ± 0.003	0.348 ± 0.009	0.324 ± 0.004
Insulin [µg/mL]	10,261 ± 35	3631 ± 13	2045 ± 17	2120 ± 12
% EE		~35		
Overall recovery %			~20	~21

* Redissolved.

## Data Availability

The original contributions presented in the study are included in the article, further inquiries can be directed to the corresponding author.

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
