# Peer review of "Archaeosomes for Oral Drug Delivery: From Continuous Microfluidics Production to Powdered Formulations"

_pharmaceutics, 2024, doi:10.3390/pharmaceutics16060694_

Round 1

Reviewer 1 Report

Comments and Suggestions for Authors

This is a very interesting paper with novel data about archeoliposomes. I think this work is very original and relevant with a clear and well-explained methodology. 

I suggest including a brief paragraph about the eventual possibility of these lipids inducing immunological responses, such as allergy, or tolerance, since they may be absorbed by Peyer's patches. I think that the negative aspects of these lipids in contact with the human body deserve to be a topic of discussion as well, in the discussion section.

Reviewer 2 Report

Comments and Suggestions for Authors

The present manuscript reports the manufacturing by microfluidic and evaluation of archaeosomes aimed for oral drug delivery. Furthermore, the products of the microfluidic process have been lyophilized or spray-dried to obtain a powder intermediate useful for preparing solid oral dosage forms. The aim of the paper does not appear very clear. The Authors stated that ‘This study aimed to establish a continuous manufacturing process of archaeosomes’ but there are not much results regarding the process. The results concern the evaluation of the quality of the archaeosomes manufactured. Therefore, in my opinion, the document should be accepted with major revisions.

The authors should underline the usefulness of archessomes: they were found to be able to protect the loaded drug and, in the meantime, the release of the drug was very slow. The Authors must indicate for which target this formulation could be useful.

The Authors should hypothesize which solid oral dosage forms should be manufactured with the lyophilized and spray-dried archaesomes powders they obtained from them.

The Authors will have to check the references of the companies: in particular whether these references are consistent with each other (the names of the cities and states are not always indicated).

The Authors must check the spaces between numbers and units of measurements.

In the Tables, all reported values must have the same number of decimals.

In the row 174, the Authors will have to specify what they mean by ‘HCl’.

The authors should explain why they performed stability studies at room temperature. These tests don't match the release tests?

The Authors would explain how they set up and optimized the microfluidic process. In particular, they did not describe how they consider the parameters they listed in the first part of Discussion chapter.

In Table 1 the difference between the second and third row preparations is not clear.

Reviewer 3 Report

Comments and Suggestions for Authors

The reviewer greatly appreciates the author's effort in shaping the manuscript. However, I have included my comments below, which I recommend for you to consider.

1.     What is the level of organic solvent in the final formulation, and how was its use in an oral dosage form justified by the authors?

2.     Line 151 mentioned the lipidic concentration in the organic solvent is 21 mg/mL. However, Line 348 shows 20 mg/mL. Please correct this.

3.     The total flow rate mentioned in the methods section (Line 162) is 1.8 mL/min, but it is 1.7 mL/min in the results section (Line 349). Please correct this.

4.     As the temperatures used were higher than the thermal stability, what is the stability of insulin in a spray-dried dosage form? Please justify.

5.     Did the authors conduct accelerated stability experiments on the developed formulations? If yes, kindly furnish the data.

Reviewer 4 Report

Comments and Suggestions for Authors

Comments on the Quality of English Language

Spelling and grammar are on par. It is a well-written paper, with only a few small technical errors.

 Please do not use the word “like” throughout the paper. Rather change to “such as”; “including”; “for example”; etc.

Round 2

Reviewer 2 Report

Comments and Suggestions for Authors

The present manuscript reports the manufacturing by microfluidic and evaluation of archaeosomes aimed for oral drug delivery. Furthermore, the products of the microfluidic process have been lyophilized or spray-dried to obtain a powder intermediate useful for preparing solid oral dosage forms. After the first revision, the aim of the paper does not appear very clear. The Authors stated that ‘This study aimed to establish a continuous manufacturing process of archaeosomes’ but there are not much results regarding the optimization of the process. In general, it does not appear that substantial changes have been introduced to the article. Therefore, in my opinion, even after the first review, the document should be accepted with major revisions.

The reviewer noted that some of the changes requested have been not done and some suggested comments have not be well understood or have not be considered. For this reason, some of those have been reproposed.

In particular, the Authors should describe how they set-up the microfluidic process and which optimization method they applied.

Then, the Authors should highlight the utility of archaesomes and emphasize whether a balance among their properties needs to be achieved: they have been demonstrated to effectively protect the loaded drug, while ensuring a slow release of the drug. The Authors should specify the intended target for which this formulation could be beneficial.

The Authors should hypothesize which solid oral dosage forms should be manufactured with the lyophilized and spray-dried archaesomes powders they obtained from them. In this regard, the Authors should consider the physico-technological properties of the obtained powders.

The Authors should explain why insulin, a peptide, has been selected as model drug. Moreover, the Authors should consider that this peptide, if released in small intestinal tract, can be metabolized, even to a large extent, by the enzymes present in the intestinal fluid and mucosa. Additionally, insulin is poorly absorbed through the membranes of the intestinal tract.

The Authors should check again if they referenced consistently the companies.

The Authors must check again if they inserted all the spaces between numbers and units of measurements.

In the Tables 2 and 3 and along the paper, the values of mean and relevant standard deviation must have the same number of decimals.

In the row 175-176, the Authors will have to specify what concentration solution of hydrochloric acid they used.

The authors should explain why they performed stability studies at room temperature. The described test seems to match the release tests?

The Authors would explain how they set up and optimized the microfluidic process. In particular, they did not describe how they consider the parameters they listed in the first part of Discussion chapter and they did not describe the trials they performed.

In the modified Table 1, it is unclear whether the third and fourth preparations, as outlined in this table, were conducted under the same conditions and using the same formulation. Could the Authors explain it in the text.

Row 687, the Authors should explain what they mean with the term ‘statistical’ in the sentence: ‘Encapsulation of hydrophilic compounds in absence of any driving force that will stimulate loading efficiency, is a statistical problem of separation between inner particle volume and total liquid preparation volume.

Round 3

Reviewer 2 Report

Comments and Suggestions for Authors

The manuscript entitled “Archaeosomes for oral drug delivery: from continuous microfluidic production to powder formulation” reports the production of archaeosomes using microfluidics. The products were then freeze-dried or spray-dried to obtain a powder intermediate intended for solid oral dosage forms. It is not clear what type of solid pharmaceutical forms could be produced because the physical-technological characteristics of the powder obtained are not described and no consideration is given regarding the dose that could be transported on the final pharmaceutical forms.

The reviewer noted that the Authors added some results regarding the process setup in the Discussion chapter that could be better explained in the Results chapter. In general, it does not appear that substantial changes have been introduced to the article. Therefore, in my opinion, even after the last review, the document should not be accepted for the publication.

The authors, then, did not highlight the usefulness of archaeosomes considering the very slow release of the loaded APIs.

The Authors did not explain why insulin, a peptide, has been selected as model drug.

The Authors should comment on the differences between the characteristics of the third and fourth preparations listed in Table 1. These should have been produced under the same process conditions and using the same formulation.

In the row 710, the Authors should explain what they mean with the term ‘statistical’ in the sentence: ‘Encapsulation of hydrophilic compounds in absence of any driving force that will stimulate loading efficiency, is a statistical problem of separation between inner particle volume and total liquid preparation volume.’

Round 4

Reviewer 2 Report

Comments and Suggestions for Authors

The manuscript entitled “Archaeosomes for oral drug delivery: from continuous microfluidic production to powder formulation” reports the production of archaeosomes using microfluidics. The products were then freeze-dried or spray-dried to obtain a powder intermediate intended for solid oral dosage forms. The Authors must consider that the physical-technological characteristics of the powder obtained could be unsuitable for the preparation of solid pharmaceutical forms, also taking into consideration the quantity of drug to be conveyed.

The reviewer noted that the authors improved the text with the aim of responding to the request posed in the third revision. Therefore, in my opinion, after the last revision, the paper should be accepted in present form.